# Genomic anatomy of male-specific microchromosomes in a gynogenetic fish

Miao Ding[1,2☯], Xi-Yin Li[1,2☯]*, Zhi-Xuan Zhu[1,2], Jun-Hui Chen[3,4], Meng Lu[1,2], Qian Shi[1,2], Yang Wang[1,2], Zhi Li[1,2], Xin Zhao[1,2], Tao Wang[1,2], Wen-Xuan Du[1,2], Chun Miao[1,2], Tian-Zi Yao[1,2], Ming-Tao Wang[1,2], Xiao-Juan Zhang[1,2], Zhong-Wei Wang[1,2], Li Zhou[1,2], Jian-Fang Gui[1,2]

**1** State Key Laboratory of Freshwater Ecology and Biotechnology, Institute of Hydrobiology, The Innovative Academy of Seed Design, Hubei Hongshan Laboratory, Chinese Academy of Sciences, Wuhan, China, **2** University of Chinese Academy of Sciences, Beijing, China, **3** BGI Genomics, BGI-Shenzhen, Shenzhen, China, **4** ShenZhen People's Hospital, Shenzhen, China

☯ These authors contributed equally to this work.
* lixiyin@ihb.ac.cn

**Data Availability Statement:** The whole genome assemblies of C. gibelio and C. auratus are deposited at GenBank database under the accession number of PRJNA546443 (BioSample SAMN11978084) and PRJNA546444 (BioSample

## Abstract

Unisexual taxa are commonly considered short-lived as the absence of meiotic recombination is supposed to accumulate deleterious mutations and hinder the creation of genetic diversity. However, the gynogenetic gibel carp (*Carassius gibelio*) with high genetic diversity and wide ecological distribution has outlived its predicted extinction time of a strict unisexual reproduction population. Unlike other unisexual vertebrates, males associated with supernumerary microchromosomes have been observed in gibel carp, which provides a unique system to explore the rationales underlying male occurrence in unisexual lineage and evolution of unisexual reproduction. Here, we identified a massively expanded satellite DNA cluster on microchromosomes of hexaploid gibel carp via comparing with the ancestral tetraploid crucian carp (*Carassius auratus*). Based on the satellite cluster, we developed a method for single chromosomal fluorescence microdissection and isolated three male-specific microchromosomes in a male metaphase cell. Genomic anatomy revealed that these male-specific microchromosomes contained homologous sequences of autosomes and abundant repetitive elements. Significantly, several potential male-specific genes with transcriptional activity were identified, among which four and five genes displayed male-specific and male-biased expression in gonads, respectively, during the developmental period of sex determination. Therefore, the male-specific microchromosomes resembling common features of sex chromosomes may be the main driving force for male occurrence in gynogenetic gibel carp, which sheds new light on the evolution of unisexual reproduction.

## Author summary

Unisexual taxa are considered short-lived as the accumulation of deleterious mutations and hindering the creation of genetic diversity. However, the gynogenetic gibel carp (*Carassius gibelio*) containing rare and variable proportions of males in wild populations has

SAMN11978330), respectively. The unassembled sequence of C. gibelio and C. auratus genome generated in this study have been submitted to the NCBI BioProject database under accession number PRJNA659101. The PacBio-SMRT data of the three male-specific microchromosomes generated in this study have been submitted to the NCBI BioProject database under accession number PRJNA658923, PRJNA658930, and PRJNA658940. The long-reads and short-reads RNA-seq data of genotypic female and male gonads at different developmental stages have been submitted to the NCBI BioProject database under accession number PRJNA658709 and PRJNA658708, respectively. The long-reads RNA-seq data of temperature-dependent female and male gonads at different developmental stages have been submitted to the NCBI BioProject database under accession number PRJNA681130. And the other data is available within the paper and its Supporting Information files.

**Funding:** This work was supported by the National Key Research and Development Project (2018YFD0900204, L.Z.), the National Natural Science Foundation of China (31873036, X.Y.L.), the Key Program of Frontier Sciences of the Chinese Academy of Sciences (QYZDY-SSW-SMC025, J.F.G.), the Strategic Priority Research Program of the Chinese Academy of Sciences (XDA24030104, J.F.G.), China Agriculture Research System of MOF and MARA (CARS-45-07, J.F.G.), the Autonomous Project of the State Key Laboratory of Freshwater Ecology and Biotechnology (2019FBZ04, J.F.G.), the Youth Innovation Promotion Association CAS (2020334, X.Y.L.). The funders had no role in study design, data collection and analysis, decision to publish, or preparation of the manuscript.

**Competing interests:** The authors have declared that no competing interests exist.

outlived its predicted time of extinction and exhibited strong environmental adaptation, which provides a special system to investigate the evolution of unisexual reproduction in vertebrates. Our previous studies have revealed that the supernumerary microchromosomes are associated with male determination in gibel carp. Here, we further isolated three male-specific supernumerary microchromosomes and revealed that they contained homologous sequences of autosomes and abundant repetitive elements. Besides, we identified several genes with transcriptional activity on these microchromosomes, especially some genes with male-specific or male-biased expression during the developmental period of sex determination. The male-specific microchromosomes with abundant repetitive elements and active male-specific/male-biased genes display common features of sex chromosomes and may be the main driving forces for male occurrence in gynogenetic gibel carp.

## Introduction

Sexual reproduction is prevalent in vertebrates, while only about 100 taxa have been documented to develop unisexual reproductive ability [1–3] since the first unisexual vertebrate Amazon molly (*Poecilia formosa*) was described in 1932 [4,5]. Unisexual vertebrates produce solely female offspring with nearly identical genetic information, mainly via three modes including parthenogenesis, gynogenesis, or hybridogenesis [2,3,6]. In parthenogenesis, females produce unreduced eggs containing the same chromosome complement as somatic cells, and these eggs develop into offspring spontaneously in the absence of males [7]. In gynogenesis, females also produce unreduced eggs with the same chromosome complement as somatic cells, but sperm are required to stimulate the eggs to initiate embryogenesis using only maternal genetic information [8]. In typical hybridogenesis, females produce reduced eggs that contain only maternal haploid chromosomes, and these eggs must be fertilized by sperm from another species. These hybridogenetic offspring contain both maternal and paternal haploid chromosomes, but only maternal haploid chromosomes remain in the reduced eggs [9].

Unisexual taxa without meiosis and meiotic recombination are supposed to be unable to purge deleterious mutations and create genetic diversity stated by Muller's ratchet, which are preconditions for adaptation to the changing environment [1,5,10,11]. Thus, unisexual lineages are considered to be short-lived, although mating costs can be avoided and high fecundity can be achieved with unisexual reproduction [3]. The predicted extinction time of a strict unisexual vertebrate population is no more than 100,000 generations [12], however, a few unisexual taxa have outlived their predicted time of extinction and exhibited strong environmental adaptation [5,13–17]. The hexaploid gibel carp (*Carassius gibelio*), which was originated from ancestral sexual tetraploid crucian carp (*Carassius auratus*) through autopolyploidy [15], can reproduce via unisexual gynogenesis using the males of sympatric host sexual species [8]. And the gynogenetic *C. gibelio* with higher genetic diversity and wider ecological distribution than its sexual progenitor *C. auratus* [15,16] has existed over 0.5 million years [15]. Besides, variable male proportions ranging from 1.2% to 26.5% have been discovered in wild populations of gynogenetic *C. gibelio* [18,19], which is unlike other unisexual taxa with all-female composition [6]. These characteristics make gynogenetic *C. gibelio* a special system to investigate the rationales underlying male occurrence in unisexual lineage and the evolution of unisexual reproduction.

Supernumerary B chromosomes, which occur in about 15% of eukaryotic species [20], are non-essential karyotypic components with non-Mendelian inheritance in addition to standard

A chromosomes (autosomes and sex chromosomes) [20–23], also known as supernumerary chromosomes, B chromosomes, or extra chromosomes. Genomic analyses have revealed that supernumerary chromosomes arise from A chromosomes and accumulate organelle genome-derived sequences [22,24]. Although supernumerary chromosomes are dispensable for the normal life of host individuals [22,25], they have been revealed to contain genes with expression activity [25–28] and be associated with some phenotypes [29]. Especially, sex-ratio distortions related with the presence of supernumerary chromosomes have been identified in many species [30–34]. In hexaploid *C. gibelio*, supernumerary microchromosomes in males are also associated with male determination [23,30], and these males with supernumerary microchromosomes contribute to the creation of genetic diversity, which is able to counter Muller's ratchet at a certain level [19,31,32]. However, the genomic components of these supernumerary microchromosomes and the underlying mechanisms of male occurrence remain elusive in gynogenetic *C. gibelio*.

In this study, we analyzed the sequence composition of three microdissected male-specific microchromosomes (MSMs) in hexaploid *C. gibelio*. These MSMs contained sequences homologous to the A chromosomes and abundant repetitive elements. Besides, several genes with transcriptional activity were identified on the MSMs, among which four and five genes showed male-specific and male-biased expression in the gonads, respectively, during the developmental period of sex determination. The features of the MSMs are similar to those of sex chromosomes, including expansion of repetitive elements and accumulation of genes with sex-specific or sex-biased expression. These results suggest that MSMs may be the main driving forces for the male occurrence in gynogenetic *C. gibelio*, which sheds new light on the evolution of unisexual reproduction.

## Results

### Expanded satellite cluster on microchromosomes

Gynogenetic hexaploid gibel carp (*C. gibelio*) was originated from ancestral sexual tetraploid crucian carp (*C. auratus*) via autopolyploidy, and gynogenetic *C. gibelio* contained several microchromosomes, which was absent in sexual *C. auratus* [15]. In order to find repetitive elements on the microchromosomes, we firstly identified the repetitive sequences in *C. gibelio* and *C. auratus* respectively, through all-to-all similarity comparison [33] using the same number of reads (1,220,000) obtained from Illumina sequencing. Subsequently, via inter-species pairwise comparative analysis, remarkable similarities in repetitive sequence composition were found between *C. gibelio* and *C. auratus* (Fig 1A). However, a bunch of repetitive sequences was found only in *C. gibelio* genome (Fig 1A), which were mainly composed of the most expanded satellite cluster (*Cg-Ca*-CL1) (Fig 1B). Besides, *Cg-Ca*-CL1 was also the most abundant repetitive sequence in *C. gibelio*, which accounted for 1.49% of whole genome size.

Intriguingly, the 137 bp consensus sequence of satellite cluster *Cg-Ca*-CL1 (S1 Table) was the same as the previously identified repeats in a male-specific sequence (*Cg*-M-s) (GenBank accession number KT260068). Five intact and four fragmental repeats of *Cg-Ca*-CL1 were distributed in *Cg*-M-s (S1A and S1B Fig). Subsequently, we performed a co-localization analysis of *Cg-Ca*-CL1 and *Cg*-M-s via fluorescence *in situ* hybridization (FISH), and found out that the FISH signals of *Cg-Ca*-CL1 resided on all microchromosomes in both female and males, which were exactly co-localized with the *Cg*-M-s signals (S2 Fig). These results indicated that the FISH signals of *Cg*-M-s might be mainly derived from these intact or fragmental repeats of *Cg-Ca*-CL1, although *Cg*-M-s has some male-specific sites other than *Cg-Ca*-CL1 repeats (S1A Fig) [23]. To reduce procedures of microchromosome identification, we designed three

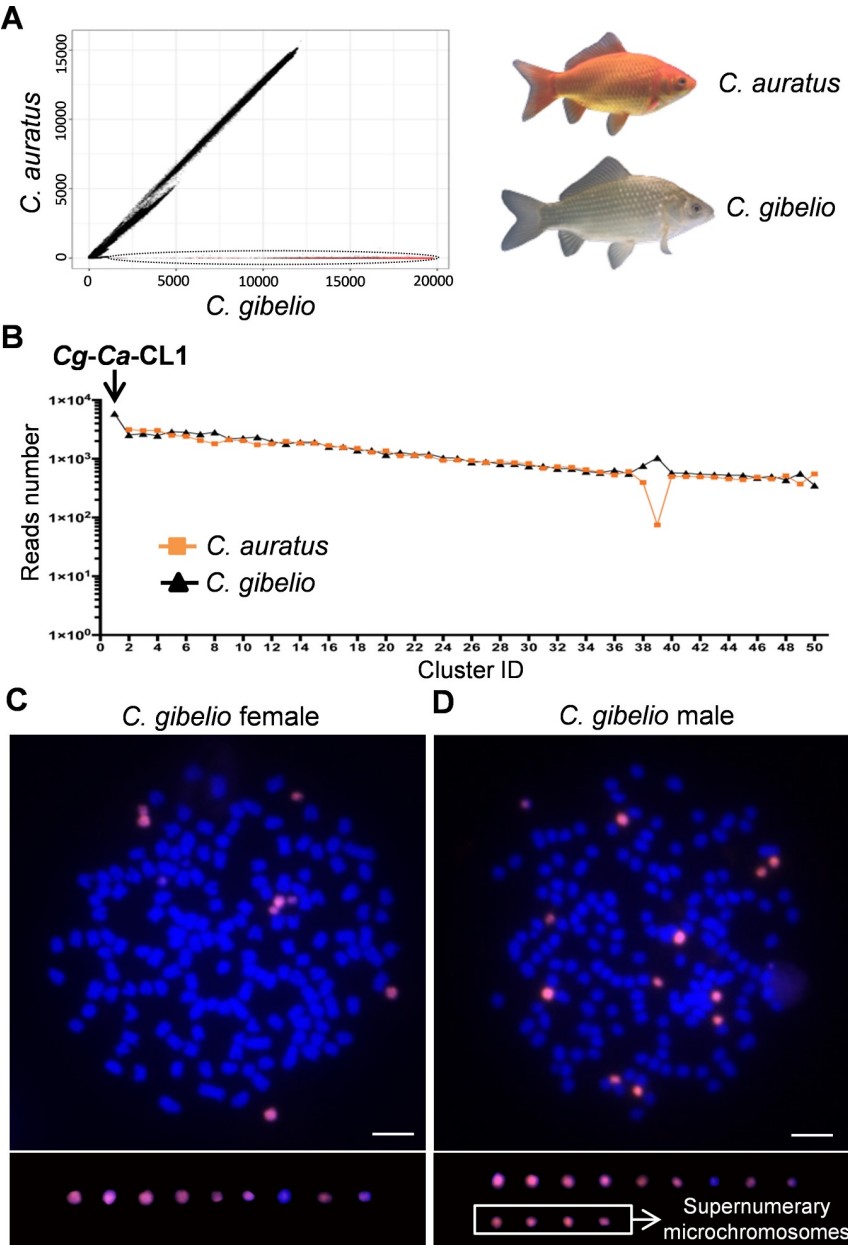

**Fig 1. Repetitive sequence expansion on microchromosomes.** (**A**) Pairwise comparison of all analyzed reads between hexaploid *C. gibelio* and tetraploid *C. auratus*. The X-axis and Y-axis show the numbers of similarity hits for each read in *C. gibelio* and *C. auratus*, respectively. Each spot corresponds to one read. The black ellipse indicates the repetitive sequences with expansion in *C. gibelio* compared to *C. auratus*. The red dots represent the reads of the most expanded satellite cluster (*Cg-Ca*-CL1). (**B**) The top 50 largest repeat clusters generated by *C. gibelio*-*C. auratus* (*Cg-Ca*) pairwise comparative analysis. The Y-axis shows the reads number in clusters, and the X-axis shows the cluster ID. (**C, D**) FISH analysis of satellite repeat cluster *Cg-Ca*-CL1 in female metaphase (C) and male metaphase (D) of *C. gibelio*. Scale bar = 5 μm. The white square indicates supernumerary microchromosomes with a male determination role in *C. gibelio*.

peptide nucleic acid (PNA) probes according to the *Cg-Ca*-CL1 satellite cluster (S1C Fig) for FISH analysis. As expected, nine and thirteen microchromosomes could be well identified by these PNA probes in females and males, respectively (Fig 1C and 1D).

*Cg-Ca*-CL1 was not detected in tetraploid *C. auratus* during repetitive sequence analysis (Fig 1A and 1B) and no positive *Cg-Ca*-CL1 signal was observed through FISH analysis (S3A and S3B Fig). However, polymerase chain reaction (PCR) showed a few low-copy amplicons in the genome of *C. auratus*, compared with abundant high-copy products in *C. gibelio* (S2C Fig). These findings suggest that the satellite cluster *Cg-Ca*-CL1 in gynogenetic hexaploid *C. gibelio* might originate from ancestral tetraploid *C. auratus* and undergo substantial expansion along with the evolution of microchromosomes after autopolyploidy [15].

## Microchromosome microdissection and male-specific microchromosome (MSM) identification

To uncover the genomic composition of MSMs, we developed a single chromosomal fluorescence microdissection technique (S4 Fig) based on FISH analysis using PNA probes and microdissection under both fluorescent and white light (see Materials and methods). We microdissected all the 13 microchromosomes from one male metaphase cell (Fig 2A) and all the 9 microchromosomes from one female metaphase cell (Fig 2B), and individually amplified these microdissected chromosomal samples via multiple displacement amplification (MDA) (Fig 2C). To examine the sex specificity of these microdissected microchromosomes, a male-specific marker derived from *Cg*-M-s (S1A and S5 Figs) was used to scan the products of each isolated microchromosome. All the microchromosomes (nine microchromosomes) from the female metaphase cell and most microchromosomes (ten microchromosomes) from the male metaphase cell had no male-specific marker, while the rest three microchromosomes from the male metaphase cell were detected to contain the male-specific marker (Fig 2D). These three microdissected microchromosomes with the male-specific marker were defined as MSMs (Fig 2D). The male metaphase cell had four extra microchromosomes than the female metaphase cell (Fig 2A and 2B), but only three extra microchromosomes in the male metaphase cell were identified as MSMs (Fig 2C and 2D). Maybe, the male metaphase has one extra microchromosome without male specificity, but we also cannot exclude the possibility that the MDA-based DNA amplification of a single microchromosome has not amplified the entire DNA.

Subsequently, the amplified DNAs of three microdissected MSMs were used as probes along with microchromosome-specific PNA probes for FISH co-localization analysis. The signals of amplified DNA were mainly localized on almost all microchromosomes in male metaphases and two microchromosomes exhibited intensive signals (Fig 2E–2G), which indicated that the technical process of single chromosomal fluorescence microdissection is accurate. Besides, some weak signals from amplified DNA were also observed in some autosomes (Fig 2E–2G), possibly due to some other repetitive elements without male specificity.

## Genomic sequences of male-specific microchromosomes

The amplified products of three microdissected MSMs were sequenced via the continuous long-read (CLR) of Sequel II (PacBio platform). After filtering low quality reads and removing adaptor sequences, a total of 6.92 Gb data were obtained from 1,545,433 clean reads with 7,974 bp N50 length. MSM 1, MSM 2, and MSM 3 generated 624,541 clean reads, 505,551 clean reads and 415,341 clean reads with data sizes of 2.56 Gb, 2.34 Gb, and 2.02 Gb, respectively (S2 Table). The clean reads of each MSM were self-corrected by CANU (S3 Table) and then assembled via CANU (S4 Table), SPAdes (S5 Table), and SMARTdenovo (S6 Table), respectively. Subsequently, the genome assembly of a female *C. gibelio* [34] and the full-length gonadal transcriptomes (S7 Table) were used as references respectively to assess the assemblies of MSMs (see Materials and methods). All the contigs and corrected reads displayed much lower alignment level to the references in sharp contrast with the clean reads (S6 Fig), which might be

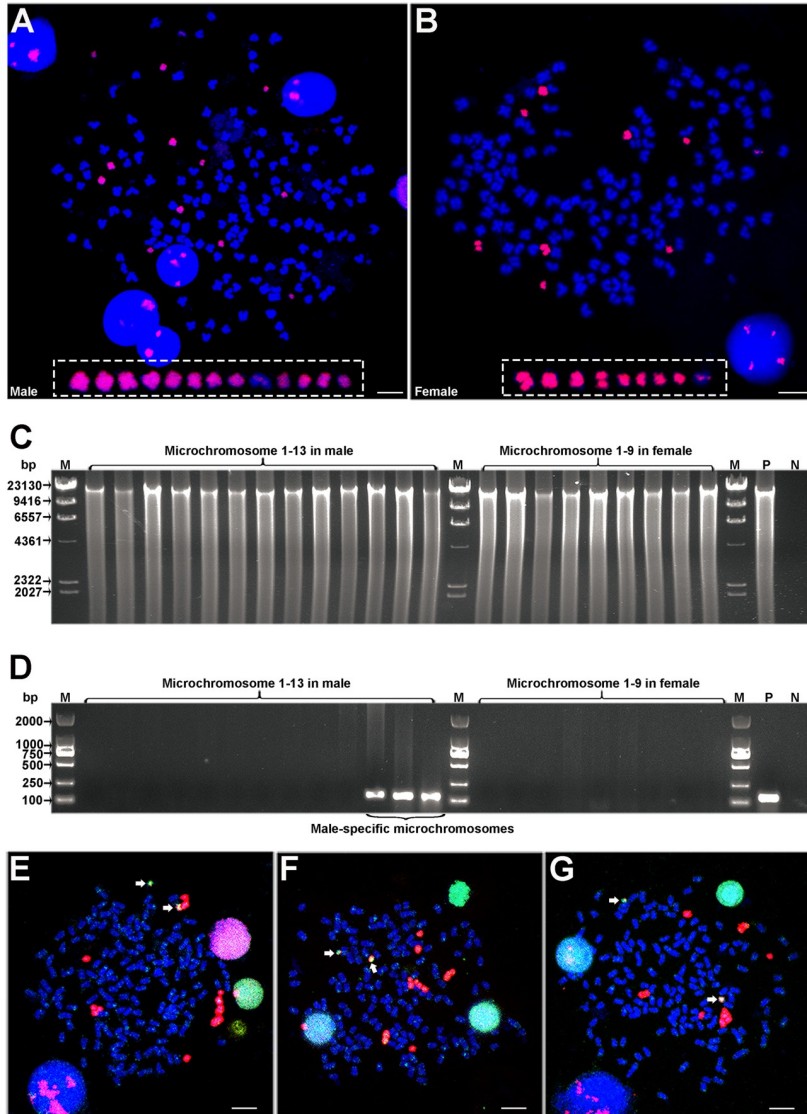

**Fig 2. Identification of three male-specific microchromosomes. (A, B)** FISH analysis on the male metaphase (A) and female metaphase (B) using PNAs as probe. The white rectangle indicates microchromosomes. Scale bar = 5 μm. (**C**) Electrophoresis of the amplified products from microdissected microchromosomes. (**D**) PCR detection of these amplified products using male-specific primers. M, marker; P, positive control using genomic DNA as template; N, negative control using water as template. (**E-G**) Co-localization of amplified DNAs and microchromosome-specific PNA probes. The amplified DNAs of MSM 1 (E), MSM 2 (F), and MSM 3 (G) were labeled with Digoxin, and the PNAs were labeled with Biotin, which appeared green and red fluorescence respectively. Scale bar = 5 μm. Mitotic cells of three males were used as replication for co-localization analysis.

caused by the removal of repetitive sequences during the process of correcting and assembling. Thus, to better reflect the reality of sequence composition, we only used the clean reads for subsequent analyses.

Mapping the MSM clean reads to the reference genome of a female *C. gibelio* [34] revealed that a total of 46.35% MSM 1 sequences, 38.84% MSM 2 sequences, and 40.01% MSM 3 sequences were homologous to the sequences from almost all linkage groups (Fig 3A–3C). With the exception of unanchored sequences, the largest number of reads from MSM 1, MSM 2, and MSM 3 were mapped to linkage groups A19, A23, and B7 respectively in *C. gibelio* (Fig

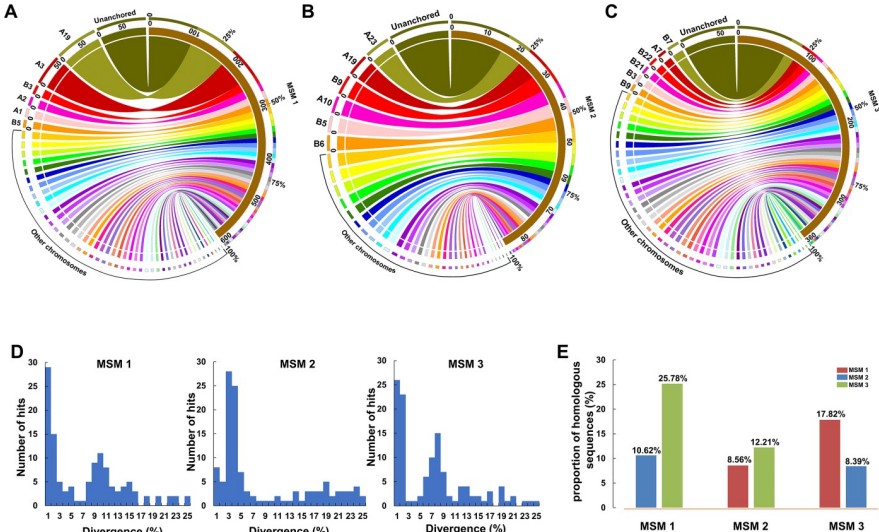

**Fig 3. Comparative genomic analysis. (A-C)** Comparative analysis between MSM sequences and the reference genome of *C. gibelio*. MSM 1 (A), MSM 2 (B), and MSM 3 (C) blocks are shown at the right and the corresponding linkage groups of reference genome are displayed at the left. (**D**) Kimura divergence between the reference genome and homologous sequences of MSMs. The X axis represents the divergence and the Y axis displays the number of hits. (E) Pairwise comparative analysis among three MSMs. The X axis indicates the compared objects and the Y axis indicates the proportion of homologous sequences.

3A–3C). Kimura distance analysis (see Materials and methods) was used to indicate the evolutionary divergence between the reference genome and homologous sequences of MSMs. Referring to the genome assembly, the aligned sequences showed two main distribution peaks in both MSM 1 (1% and 10% divergence) and MSM 3 (1% and 8% divergence), while the aligned sequences of MSM 2 were concentrated in only one main peak with 3% divergence (Fig 3D). Besides, pairwise comparative analysis revealed that the proportion of homologous sequences between MSM 1 and MSM 3 is over 2-fold higher than that between MSM 2 and MSM 1 or between MSM 2 and MSM 3 (Fig 3E).

## Characterization of repetitive elements on male-specific microchromosomes

To search repetitive elements, we first constructed a TE database specific to the hexaploid *C. gibelio* via combined prediction of the signature and homology (see Materials and methods). Afterward, the constructed *C. gibelio*-specific TE database and the known metazoan repetitive database (RepBase 23.07) were used to identify repetitive elements on MSMs. We found that 89.40%, 85.76%, and 93.58% of reads contained at least one repetitive element on MSM 1, MSM 2, and MSM 3, respectively (Fig 4A). Further, 52.00%, 62.57%, and 63.49% of the sequences of MSM 1, MSM 2, and MSM 3 were identified as repetitive elements including satellite, TE, and unknown repeats (Fig 4B), which were 1.2- to 1.5-fold higher than repeat content in the whole genome assembly of a female individual containing microchromosomes (42.6%) [34]. Among the three MSMs, Y-chromosome satellites were the most abundant satellite sequences (Fig 4C), and DNA transposons were the most frequent repeats of TEs (Fig 4D). Besides, most of the TE families on these MSMs amplified to a medium (11–100 copies) or high (>100 copies) copy number (Fig 4E). These results indicate that numerous repetitive elements have been accumulated on the MSMs during the evolutionary process.

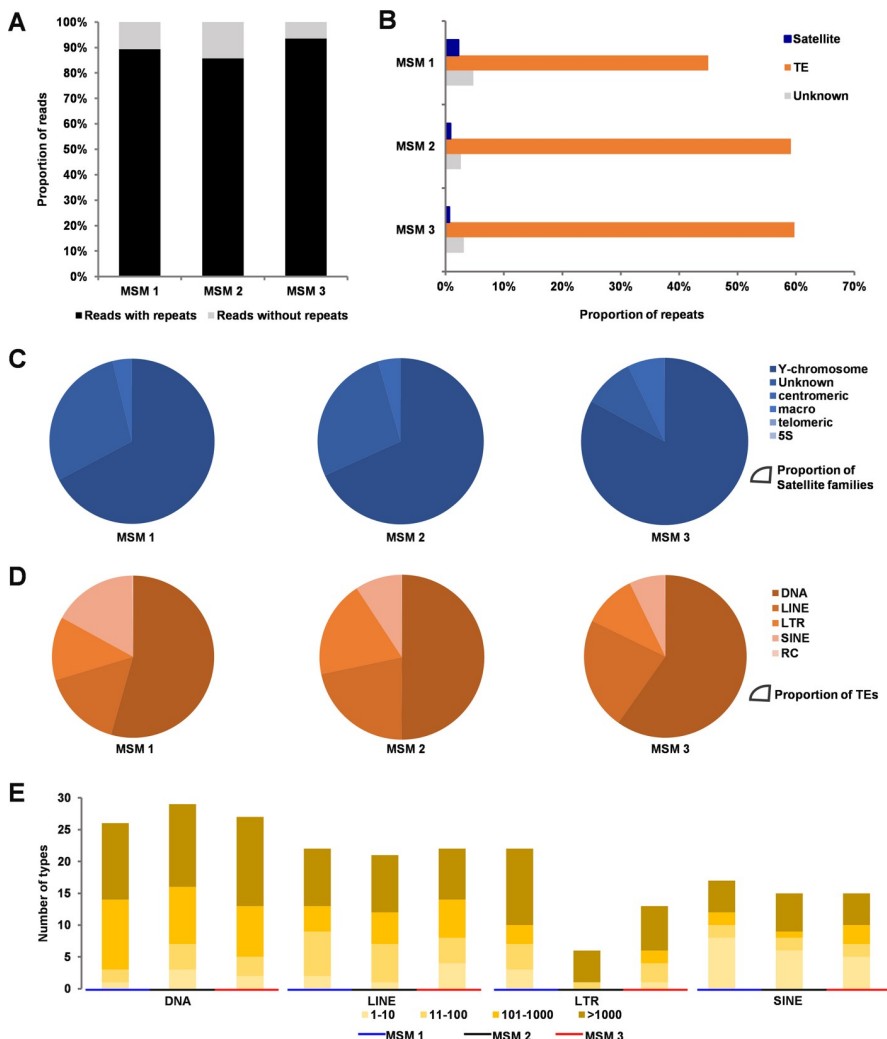

**Fig 4. Analysis of repetitive elements.** (**A**) The proportion of sequencing reads containing repetitive elements. The Y axis indicates the proportion of reads. (**B**) The proportion of repetitive elements including satellite, TE, and unknown. (**C, D**) Types and proportion of satellite repeats (C) and TEs (D). (**E**) Copy numbers of different types of TEs.

## Identification of genes derived from male-specific microchromosomes

We performed gene annotation of MSMs via Minimap2 and the Basic Local Alignment Search Tool (BLAST) using gonadal transcriptomes at ten developmental stages of hexaploid *C. gibelio* (Fig 5A and S7 Table) and the coding sequences of nine fish species (Fig 5B) as references. A total of 487 genes (228, 88, and 203 on MSM 1, MSM 2, and MSM 3, respectively) were identified on MSMs, among which 2 genes were shared by all the three MSMs, and 25 genes were shared by MSM 1 and MSM 3 (Fig 5C). Meanwhile, MSM 1 and MSM 2 shared only 1 gene, and MSM 2 and MSM 3 shared 2 genes (Fig 5C). A total of 1.45%, 0.02%, and 0.51% clean reads were revealed to contain gene sequences on MSM 1, MSM 2, and MSM 3, respectively (S7 Fig). And these identified gene sequences account for 1.067%, 0.017%, and 0.439% total sequences of MSM 1, MSM 2, and MSM 3, respectively. Subsequently, we performed analysis of gene integrity and found out that the majority annotated genes had the integrity scores less than 10%, which were accounted for 78.95% (180 genes), 81.82% (72 genes), and 76.85% (156 genes) of annotated genes in MSM 1, MSM 2, and MSM 3, respectively. Meanwhile, there are

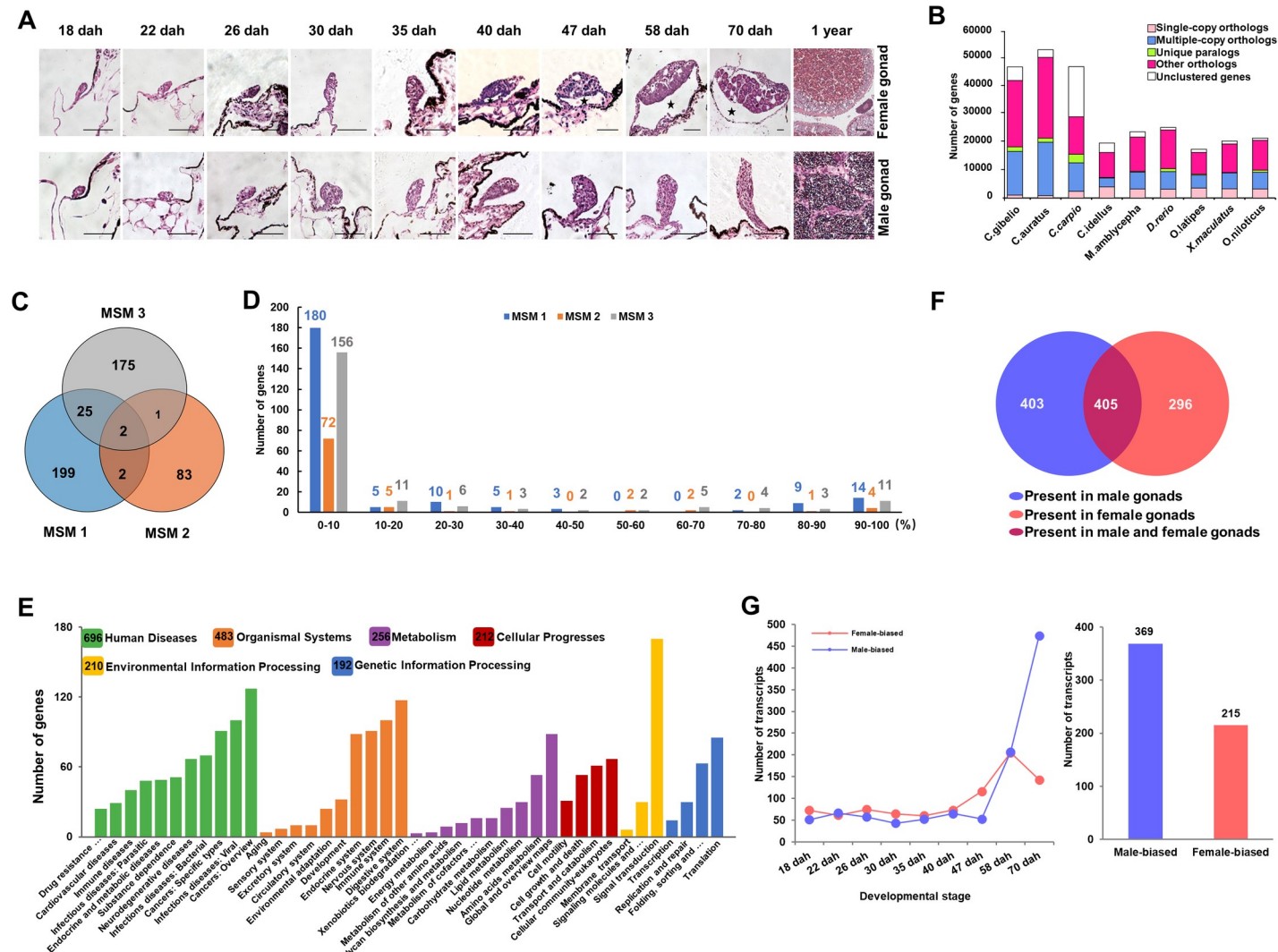

**Fig 5. Identification of genes and the corresponding transcripts.** (**A**) The hematoxylin-eosin staining of genotypic female and male gonads at 10 developmental stages. Scale bar = 50 μm. (**B**) The distribution of different types of gene families among nine species. The X axis shows nine species and the Y axis indicates the number of genes. (**C**) Numbers of identified genes on MSM 1, MSM 2 and MSM 3. (**D**) Results of gene integrity analysis. X axis indicates each integrity percentage bins and Y axis shows the number of genes. (**E**) KEGG analysis of the MSM-linked transcripts. (**F**) The distribution of MSM-linked transcripts in male and female gonads. (**G**) Sex-biased transcripts during the early stages of gonadal development. Dah, days after hatch.

only 10.96% (25 genes), 10.23% (9 genes), and 12.32% (25 genes) annotated genes with integrity score over 50% in MSM 1, MSM 2, and MSM 3, respectively (Fig 5D). All the annotated 487 genes corresponded to 1104 transcripts (MSM-linked transcripts), which belonged to six Kyoto Encyclopedia of Genes and Genomes (KEGG) pathways related to "Human Disease" (696 transcripts), "Organismal Systems" (483 transcripts), "Metabolism" (256 transcripts), "Cellular Progresses" (212 transcripts), "Environmental Information Processing" (210 transcripts) and "Genetic Information Processing" (192 transcripts) (Fig 5E). Among these 1104 MSM-linked transcripts, 403 and 296 transcripts were only detected in the transcriptome of male and female gonads, respectively (Fig 5F). Further, sex-biased transcripts were identified via differential expression analysis (false discovery rate < 0.05, |log$_2$FoldChange| >1) at each developmental stage. A total of 369 male-biased transcripts and 215 female-biased transcripts

were identified from nine developmental stages, specifically at 18, 22, 26, 30, 35, 40, 47, 58, and 70 days after hatch (dah) (Fig 5G).

## Gene fragments with male-specific or male-biased expression

To identify MSM genes with male-specific expression, 808 MSM-linked full-length transcripts presenting in male gonads (Fig 5F) were selected for subsequent bioinformatic subtraction (see Materials and methods). Firstly, two transcriptomes derived from individuals without MSMs were used for transcriptome subtraction, and the 334 transcripts that had high identity with these two transcriptomes were excluded (Fig 6A). After transcriptome subtraction, the remaining 474 transcripts were then mapped to the reference genome of a female *C. gibelio* [34] for genome subtraction. And then 125 transcripts that had high identity with the female genome were excluded and 349 transcripts were obtained after genome subtraction (Fig 6B). Commonly the full-length transcripts cannot continuously be aligned to the sequences of MSMs, not only because of the intron sequences on MSMs but also as many genes have been duplicated on MSMs as partial truncated genes [20,28,35–37]. Thus, we mapped the remaining 349 transcripts to the sequences of MSMs via BLAST with default parameter, and 10,213 well-aligned sequences of MSMs (identity>75% and aligned length $\geq$ 200 bp) were defined as gene fragments. To be more specific, the second round of transcriptome/genome subtractions was performed on these gene fragments, and 8,599 gene fragments were excluded. Finally, 1,614 gene fragments remained as the potential male-specific gene fragments on MSMs (Fig 6C).

To identify the genes crucial for sex determination, we screened the gonadal expression of these 1,614 potential male-specific gene fragments on MSMs before gonadal morphological differentiation (40 dah) (Fig 5A), via Illumina sequencing data analysis (S8 Table). We found out that a total of 159 gene fragments had higher transcription level in the male gonads than in the female gonads at least at one gonadal developmental stage before 40 dah (38, 42, 43, 35, and 60 gene fragments with higher transcriptional expression in male gonads than in the female gonads at 18, 22, 26, 30, and 35 dah, respectively) (Fig 6D and 6E). After manual subtraction of the highly overlapped fragments via pairwise sequence alignment, only 42 unique gene fragments remained, and 10 gene fragments were identified to have a conserved coding sequence compared to their homolog in A chromosomes of *C. gibelio* or other species (S9 Table).

Subsequently, relative quantitative real-time PCR was performed to analyze the gonadal expression patterns of all the 42 gene fragments before gonadal morphological differentiation, which was the developmental period of sex determination (Figs 6F and S8). At last, four (*trpv4*, *arih2*, *trim16l*, and *capb5b*) and five (*pex11b*, *tmem183a*, *gabrb3*, *pnn*, and *dcbld1l*) gene fragments were confirmed to display male-specific and male-biased expression respectively (Fig 6F), among which two gene fragments (*trim16l* and *tmem183a*) contained a potential coding sequence. Thus, these MSMs containing genes with male-specific or male-biased expression might be beneficial for male occurrence.

## Discussion

The consequences of polyploidy are frequently related to unisexual reproduction modes such as gynogenesis, parthenogenesis, hybridogenesis, and kleptogenesis [2,38–40]. Hexaploid *C. gibelio* was originated from ancestral tetraploid *C. auratus* at about 0.5 Mya via autopolyploidy [15,16,41], and the newly formed hexaploid *C. gibelio* broke through the reproduction bottleneck via unisexual gynogenesis [8,42]. Although gynogenesis has the ability to avoid mating costs and obtain high fecundity, gynogenetic taxa without meiosis cannot purge deleterious mutation and create genetic diversity, which will lead to the eventual extinction stated by Muller's ratchet [1–3,10,11]. However, the gynogenetic *C. gibelio* has higher genetic diversity and

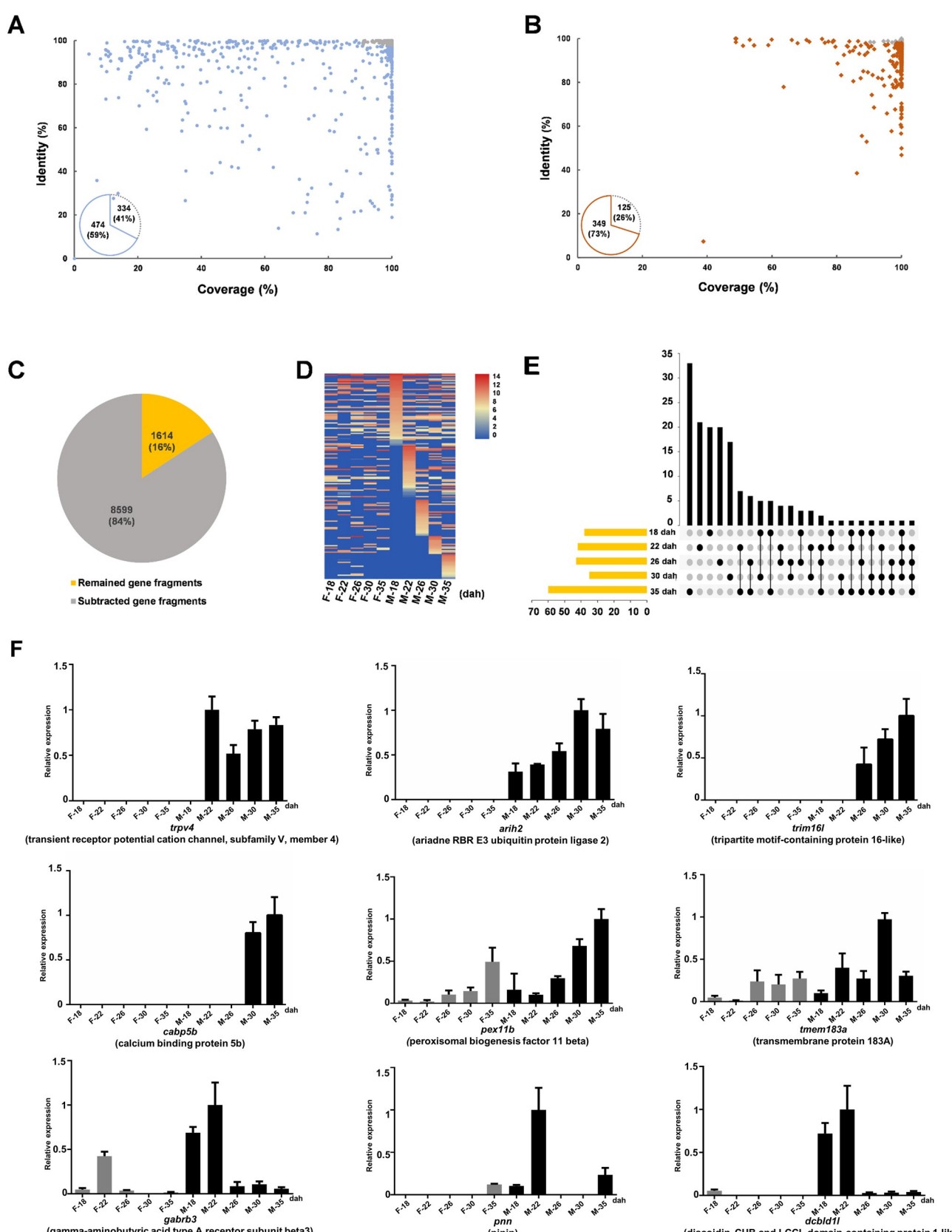

**Fig 6. Potential male-specific gene fragments with transcriptional activity. (A, B)** Transcriptome subtraction (A) and subsequent genome subtraction (B) performed on MSM-linked transcripts. The grey dots represent transcripts with the coverage > 90% and identity > 98%. The pie charts indicate the number of discarded (grey) and remained transcripts (wathet and brown) **(C)** The pie chart of the second round of transcriptome/genome subtractions performed on gene fragments. **(D)** Heatmap of the gene fragments with higher expression in male gonads than in female gonads at least at one developmental stage from 18 to 35 dah. **(E)** The distribution of the gene fragments with higher expression in male gonads than in female gonads. The axis indicates the number of gene fragments. **(F)** qPCR detection of gene fragments with male-specific or male-biased expression at early gonadal developmental stages including 18, 22, 26, 30, and 35 dah. The X axis represents the stages of gonad development. The Y axis represents the relative expression, and the highest expression level of each gene fragment was used as control and defined as 1. F, female; M, male.

wider geographic distributions than its sexual progenitor *C. auratus* [16,19]. Besides, given a generation time of 1–2 years, the gynogenetic *C. gibelio* has existed for about 250,000–500,000 generations [15], which has exceeded the predicted extinction generation of a strict unisexual reproduction population (100,000 generations) [5,12]. In contrast with other unisexual vertebrates composed of all females, males containing MSMs have been observed in wild populations and artificially propagated strains of *C. gibelio* [18,23]. These males with MSMs can initiate a variant of gynogenesis along with male occurrence, accumulation of microchromosomes, and creation of genetic diversity in the offspring [31,32], which can contribute to the environmental adaptation and evolutionary long existence of gynogenetic *C. gibelio*.

The close association between the presence of supernumerary chromosomes and sex-ratio distortion is not only observed in *C. gibelio* [23,42], but also has been demonstrated in many other taxa [43–46]. Besides, the supernumerary chromosomes have also been revealed to have a functional effect on sex determination in *Lithochromis rubripinnis* [35], *Nasonia vitripennis*, and *Trichogramma kaykai* [47,48]. In *C. gibelio*, male-specific supernumerary microchromosomes accumulate numerous repetitive elements (Figs 1 and 4) and contain active genes with male-specific or male-biased expression during the developmental period of sex determination, which are usually accompanied with the evolution of sex chromosomes [49–53]. Therefore, we could deduce that the MSMs resembling common features of sex chromosomes may be the main driving force for male occurrence in the gynogenetic *C. gibelio*.

There are two possibilities for the origin of MSMs (Fig 7). The first one is that the MSMs might emerge along with autopolyploidy at about 0.5 Mya. These microchromosomes acquired the sex-determining gene/genes from the A chromosomes of sexual progenitor during autopolyploidy and male *C. gibelio* emerged at the beginning of hexaploid *C. gibelio* formation. The second possibility is that the MSMs might form during the evolutionary process after autopolyploidy, in which no male *C. gibelio* emerged at the beginning of hexaploid *C. gibelio* formation. The sex-determining gene/genes were acquired on MSMs during the evolutionary trajectory of gynogenesis. And the sex determinant might be derived from the duplicates of A chromosomes of gynogenetic *C. gibelio* or the DNA introgression of host sexual species. Thus, further identification of sex-determining gene/genes on MSMs is essential for unveiling the origin and evolution of MSMs in the gynogenetic *C. gibelio*.

Although supernumerary chromosomes are nonessential genetic elements, many supernumerary chromosomes have the intrinsic ability to transmit themselves at frequencies above that predicted by Mendelian rules [54,55]. Similarly, the microchromosomes derived from genotypic males of *C. gibelio* also can be accumulated in the offspring [18,31]. So there is a possibility that the MSMs may be transmitted steadily across generations as a selfish genomic parasite, and the gynogenetic *C. gibelio* will maintain the present status with MSMs for male determination. Meanwhile, we also cannot exclude another possibility that the MSMs may evolve to sex chromosome or provide materials for sex chromosome evolution, as sex chromosomes in some species have been demonstrated to be evolved from supernumerary chromosomes, such as the Y chromosome in *Drosophila* species [56,57], W chromosome in Lepidoptera [58], and sex chromosomes in some cichlid fish species [46,59].

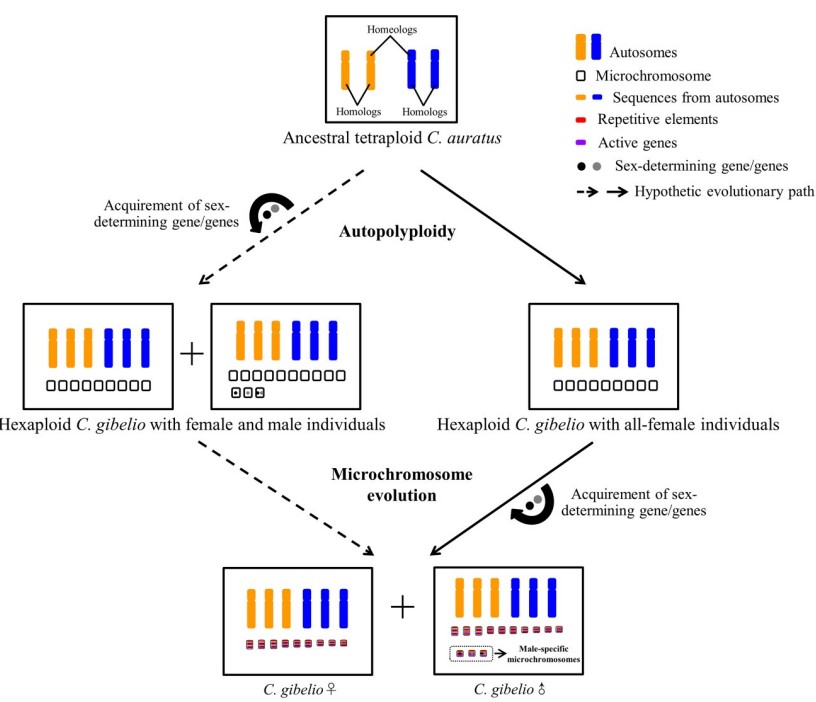

**Fig 7. Schematic diagram of MSM origin and male occurrence in *C. gibelio*.** Hexaploid *C. gibelio* was originated from ancestral tetraploid *C. auratus* via autopolyploidy, and the newly formed hexaploid *C. gibelio* reproduced via unisexual gynogenesis. Two possible evolutionary trajectories of MSM origin and male occurrence in *C. gibelio* were indicated by the dashed arrow and solid arrow respectively. One possibility is that MSMs and males might emerge at the beginning of *C. gibelio* formation via autopolyploidy, and the sex-determining gene/genes might be accumulated from the A chromosomes of sexual progenitor during autopolyploidy (dashed arrow). The other possibility is that MSMs and males did not emerge at the beginning of *C. gibelio* formation but formed during the evolutionary process after autopolyploidy. And the sex-determining gene/genes might be acquired from the duplicates of A chromosomes of *C. gibelio* or the DNA introgression of host sexual species (solid arrow).

In conclusion, we isolated three MSMs in gynogenetic *C. gibelio* and characterized abundant repetitive elements and some genes with male-specific/male-biased expression on MSMs. Our present results indicate that MSMs could be responsible for male occurrence in *C. gibelio* [18,23,31], which can facilitate further identification of the sex-determining gene/genes on MSMs. Besides, continued investigations on sex determination mechanism and reproduction mode of the gynogenetic *C. gibelio* with male occurrence will provide insights into the evolution of unisexual reproduction.

## Materials and methods

### Ethics statement

Animal experiments and treatments were performed according to the Guide for Animal Care and Use Committee of Institute of Hydrobiology, Chinese Academy of Sciences (IHB, CAS, Protocol No. 2016–018).

### Experimental fish

Experimental fish species including hexaploid gibel carp (*C. gibelio*), tetraploid crucian carp (*C. auratus*), and red common carp (*Cyprinus carpio*) were provided by the National Aquatic

Biological Resource Center (NABRC), Institute of Hydrobiology, Chinese Academy of Sciences, Wuhan, China.

## Illumina sequencing and identification of repetitive sequences

One female *C. auratus* and one female *C. gibelio* were used for re-sequencing, which was performed by Illumina (Hiseq X-ten) platform and yielded quality filtered paired-end reads with a length of 150 bp. The same amount of reads (1,220,000) were randomly selected from each fish sample for pairwise comparison to identify the repetitive sequences. Repetitive sequences were analyzed by all-to-all similarity comparisons and graph-based clustering at the RepeatExplorer platform with default parameters on a Galaxy server [33].

## Fluorescence *in situ* hybridization (FISH)

Chromosome preparation was performed as previously described [60]. Repetitive sequence cluster *Cg-Ca*-CL1 was labeled by Biotin-Nick Translation Mix (Roche) and used as the probe. The male-specific sequence (*Cg*-M-s) (GenBank accession number KT260068) and the amplified products of microdissected microchromosome were labeled by DIG-Nick Translation Mix (Roche) and used as the probe. FISH analysis was performed as previously described [61].

## Single chromosomal fluorescence microdissection

One male individual and one female individual were used for single chromosomal fluorescence microdissection. Three PNA probes, which were designed according to the repetitive sequence cluster *Cg-Ca*-CL1 and labeled with Cy3 (S1 Fig), were used for FISH analysis with some modifications. The 60 µl hybridization mix contained 50% deionized formamide, 0.5 µg/µl sheared salmon sperm DNA, 0.1% SDS, 20×SSC, 20% dextran sulphate, and 3 µl PNA probes (each probe 1 µl, 50 ng/µl). The mixture was denatured at 73°C for 3.5 min and immediately transferred into ice. After cooling, the hybridization mix was placed on the slides without any bubble and kept at 37°C overnight for hybridization. Single chromosome microdissection was performed as described previously [62] with minor modifications. All the equipment, microneedles, and RNase/DNase-free tubes were treated with UV irradiation for 30 min to decrease the DNA contaminant [63]. The single microchromosome was scraped from metaphase after FISH analysis with a glass microneedle under both fluorescent and white light, using the Eppendorf TransferMan 4 micromanipulator and Nikon Ti-E microscope. The scraped microchromosome was transferred into a 200 µl RNase/DNase-free tube on ice immediately.

We microdissected all the 13 microchromosomes from one male metaphase cell and all the 9 microchromosomes from one female metaphase cell. After all the microchromosomes were isolated respectively, the microdissected microchromosomes were individually amplified via multiple displacement amplification (MDA) using the REPLI-g Single Cell Kit (Qiagen), following the manufacturer's protocol [64]. To eliminate the contamination of extraneous DNA, all the operations were performed in a vertical clean bench, and all the instruments were treated with UV irradiation for 30 min. RNase/DNase-free water and 25 pg genomic DNA were used as negative and positive control respectively for amplification in vitro. The amplified products were analyzed by 1% agarose gel electrophoresis and then purified via the AxyPrep PCR Cleanup Kit (Axygen) with 30 µl eluent, whose concentration was measured using Nano-Drop 2000 (Thermo). And all the products were stored at -20°C until use. Mitotic cells of three males were used as replication for co-localization of amplified DNAs and microchromosome-specific PNA probes.

## Sequencing and assembling of MSMs

The MDA products of three microdissected MSMs were purified and then used to construct libraries with 4.5–10 kb inserts following the protocol of the PacBio template preparation kit. CLR technology was used and sequencing reactions were performed on the PacBio Sequel II platform instrument with Sequel Sequencing Kit 2.1 (Pacific Biosciences) and Sequel SMRT Cell 1M v2 Tray, at BGI-shenzhen, China. After filtering low quality reads and removing adaptor sequences, the remained clean reads were self-corrected using CANU [65] (version 1.4) with parameters genomeSize = 20m, errorRate = 0.013, corOutCoverage = 60, corMhapSensitivity = normal, min-ReadLength = 500, and corMinEvidenceLength = 500. And the corrected reads were subjected to three widely-used PacBio assembler, including CANU (version 1.4) with default parameters, SPAdes (SPAdes-3.12.0) with default parameters, and SMARTdenovo (smartdenovo-170825) with parameters of -S 4, -k 21, -z 12, -Z 300, -U, -1, -m 0.6, -A 1000. And all the contigs derived from these three methods were polished by Illumina short reads using Pilon program [66] (version:2.3).

To assess the assemblies of MSMs, two databases were used as references respectively including the genome assembly of a female *C. gibelio* [34] and the full-length gonadal transcriptomes of genotypic females and males at ten developmental stages (S7 Table). The clean reads, corrected reads, CANU contigs, SPAdes contigs, and SMARTdenovo contigs of MSMs were mapped to the two references respectively, via BLAST (default parameters). Subsequently, the aligned hits (identity >75% and aligned length ≥ 100 bp) were used to generate genome blocks and transcriptome blocks using our custom python script.

## Production of genotypic male offspring and RNA-seq

To obtain a high proportion of genotypic male offspring in *C. gibelio*, we firstly produced sex-reversed physiological females from the genotypic males (with MSMs) via estradiol treatment [23]. Subsequently, the sex-reversed physiological female (with MSMs) was mated with a normal genotypic male (with MSMs), and the proportion of genotypic males in the offspring was 96.1% (S9A Fig). Meanwhile, we also reproduce all-female offspring via unisexual gynogenesis as the previous description, that the ovulated eggs from a female *C. gibelio* were inseminated with the sperm from another species red common carp (S9B Fig) [23]. All the lava in these two families were reared at normal water temperature about 20°C (±1°C), and these two families were constructed in the strain DA of *C. gibelio* as the sex of individuals in strain DA could not be easily affected by rearing temperature [18].

Genotypic female gonads were obtained from the gynogenetic family with all-female offspring (S9B Fig), while genotypic male gonads were obtained from the family with a high proportion of male offspring (S9A Fig), in which female individuals were excluded via analysis of male-specific marker identified previously [23]. Gonads at ten developmental stages from 198 genotypic females and 198 genotypic males were sampled for subsequent morphological observation and transcriptome analysis. And a total of 30, 30, 20, 20, 20, 20, 10, 10, 5, and 3 gonads were pooled for RNA isolation at the stage of 18, 22, 26, 30, 35, 40, 47, 58, 70 dah, and mature gonads with 1 year old respectively. And 3 female gonads and 3 male gonads were sampled at each stage for morphological observation.

Ten RNA samples of genotypic female gonads at different stages were pooled into one female sample, and ten RNA samples of genotypic male gonads at different stages were also pooled into one male sample. These two pooled samples were sequenced on the PacBio Sequel platform respectively (S7 Table). IsoSeq v3 (https://github.com/ben-lerch/IsoSeq-3.0) was used to cluster and polish isoforms with default parameters. TransDecoder (https://github.com/TransDecoder/TransDecoder/wiki) was used to identify candidate coding regions within the final polished isoforms.

Moreover, nine RNA samples of genotypic female gonads and nine RNA samples of genotypic male gonads at 18, 22, 26, 30, 35, 40, 47, 58, and 70 dah were sequenced respectively via Illumina Hiseq platform at BGI-shenzhen, China (S8 Table). Illumina clean reads were mapped to PacBio full-length transcriptome, and Fragments per Kilobase Million (FPKM) was used to quantify gene expression, which was calculated via RSEM (Version 1.2.12) [67,68]. To normalize gene expression, if the FPKM was 0, it would be modified to 0.001.

## Production of temperature-dependent male offspring and RNA-seq

Genotypic sex determination (GSD) and temperature-dependent sex determination (TSD) were revealed to coexist in the hexaploid *C. gibelio* [18,23,42]. The temperature-dependent males, which were determined by the ambient temperature during larval development, contained no MSMs. To obtain temperature-dependent all-male offspring without MSMs, the gynogenetic larvae were raised at 32˚C (±1˚C) since first feeding for 30 days as previously reported [18,32]. Meanwhile, the gynogenetic larvae from the same family were raised at normal temperature 20˚C (±1˚C), which generated all-female offspring. The temperature-dependent all-male offspring and the corresponding all-female offspring were constructed in the strain A+ of *C. gibelio*, as the sex of individuals in strain A+ could be easily affected by rearing temperature [18].

Gonads sampled from 93 temperature-dependent males and 93 females were used for transcriptome analysis. And a total of 30, 30, 20, 10, 3 gonads were pooled for RNA isolation at the stage of 6 dah, 16 dah, 30 dah, 60 dah, and mature gonads with 1 year old respectively. The RNA samples of temperature-dependent males and the corresponding females were pooled into one sample, and sequenced on the PacBio Sequel platform (S10 Table).

## Comparative analysis

Clean reads of MSMs were mapped against the female *C. gibelio* genome (GenBank assembly accession: PRJNA546443), using Minimap2 with default parameters [69]. Self-python was used to extract matched sequences (MapQ >10), which were merged to blocks according to overlap region. The alignments between the identified blocks of each MSM and female *C. gibelio* genome were displayed by the Circos software (version: 0.69–6) [70]. Kimura distance analysis [71] was used to infer the sequence divergence between the MSMs and A chromosome sequence, based on calculating the pairwise divergence of the aligned hits. The aligned hits of each MSM were randomly selected in different length sections, and a total of 120 aligned hits were selected for each MSM. The Kimura value of each aligned hits were calculated by MEGA [72]. Pairwise comparisons among these three MSMs were performed by BLAST with default parameters.

## Construction of transposable element database and identification of repetitive elements

The assembly of *C. gibelio* genome (GenBank accession: PRJNA546443) was used to construct a specific transposable element (TE) database for hexaploid *C. gibelio* by a combinatory prediction of signature and homology as follows. During the process of the signature-based prediction, several programs were implemented for given classes of TEs with particular features. Full-length LTR retrotransposons were identified according to Ray et al. [73], with modifications. The candidate LTR elements were ultimately aligned with sequence identity parameters 0.9 by CD-HIT-EST [74] in order to reduce the redundancy. Helitron transposons were collected by HelitronScanner based on a two-layer local combinational variable (LCV) algorithm [75]. The input genomes were scanned using LCVs that were extracted by known Helitrons,

and putative Helitrons were drawn using the parameters "-ht 5 -tt 10" to avoid high false positives and missing true Helitrons. Non-autonomous DNA transposons were identified using MITE-Hunter [76] with the "-P 0.1" parameters. Autonomous non-LTR transposons were identified and classified using MGEScan-non-LTR with default parameters [77]. SINE transposons were predicted on the basis of structural features by using SINE-Finder [78], and then aligned using Needle in EMBOSS package (version 6.6.0) [79], with the output used for assigning the SINE elements to respective families if they shared 60% (or more) similarities.

The TEs identified from the way of signature-based prediction were then used to mask *C. gibelio* genome by RepeatMasker (http://www.repeatmasker.org, version 4.0.7) with WUBlast engine. The unmasked portion of genomes was subsequently used for homology-based prediction. Superfamilies of the putative TE elements were classified according to the conserved domains or the sequences of terminal invert repeats (TIR) and target site duplication (TSD). The open reading frame (ORF) within the full length LTR element, which was obtained by EMBOSS Getorf program [79], was used to search for known coding domains (e.g. Gag, Protease (PR), Reverse Transcriptase (RT), Ribonuclease H (RN), Integrase (INT), Envelope (ENV), Transposase (TR), etc.), through HMMER (version 3.1b) [80] with pHMMs model downloaded from GyDB (Gypsy database 2.0) [81]. Classification of LTR elements with complete gag-pol structures was based on the order of RT and INT, which was the distinction of two main LTR superfamilies *Gypsy* and *Copia* [82]. The rest of LTR elements were categorized into *LARDs* (> 4 kb) and *TRIMs* (< 4 kb). A homology search of The MITE elements against RepBase (version 23.07) was performed by RepeatMasker. TIRs and TSDs identified by MITE-Hunter were compared with the previously characterized TIRs and TSDs in plants and animals [83–85] through a custom Perl script. The output of RepeatMasker and TIR-TSD searching were the bases of MITEs classification. The unmasked MITEs with ambiguous (or unknown) TIRs and TSDs will be classified as unknowns. Classification of TEs identified by RepeatModeler was performed as described previously [73].

The constructed *C. gibelio*-specific TE database and the known metazoan repetitive database (RepBase 23.06) were used to identify repetitive elements on MSMs, using RepeatMasker (version: version open-4.0.6) with parameters -nolow -no_is -norna -engine ncbi -parallel 1.

## Identification of genes on MSMs and their corresponding transcripts

To screen the MSMs-linked genes, two databases including gonadal full-length transcriptomes at ten developmental stages of hexaploid *C. gibelio* (S7 Table) and the coding sequence (CDS) pool of nine fish species were used as references. The coding sequence pool of nine species consisted of *C. gibelio* (GenBank accession: PRJNA546443), *C. auratus* (GenBank accession: PRJNA546444), *C. carpio* (http://www.carpbase.org/download_home.php), *Ctenopharyngodon idellus* (*C. idellus*) (http://www.ncgr.ac.cn/grasscarp/), *Megalobrama amblycephala* (*M. amblycephala*) (http://bream.hzau.edu.cn/page/species/download.html#1), *Danio rerio* (*D. rerio*) (ftp://ftp.ensembl.org/pub/release-96/fasta/danio_rerio/), *Oryzias latipes* (*O. latipes*) (ftp://ftp.ensembl.org/pub/release-96/fasta/oryzias_latipes/), *Xiphophorus maculatus* (*X. maculatus*) (ftp://ftp.ensembl.org/pub/release-96/fasta/xiphophorus_maculatus/) and *Oreochromis niloticus* (*O. niloticus*) (ftp://ftp.ensembl.org/pub/release-96/fasta/oreochromis_niloticus/), which were clustered using OrthoMCL based on an all-to-all BLASTP strategy with the default parameters [86,87]. The sequences of MSMs were mapped to the references via Minimap2 (MapQ ≥ 10 and aligned length ≥ 200 bp) and BLAST (identity >75% and aligned length ≥ 200 bp).

To identify the corresponding transcripts of genes identified by the CDS pool, the reference CDSs were mapped against the gonadal transcriptomes of *C. gibelio* (Fig 5A and S7 Table) via

BLAST. And the following criteria of BLAST were applied that cumulative identity percentage (CIP) $\geq$ 60% and cumulative alignment length percentage (CALP) $\geq$ 70%. CIP corresponds to the cumulative percentage of sequence identity obtained for all of the high scoring pairs (HSPs) (CIP = [$\Sigma$ ID by HSP/AL] $^*$100). CALP represents the sum of the HSP aligned length (AL) for all of the HSPs divided by the length of the query sequence (CALP = $\Sigma$AL/query length).

TransDecoder (https://github.com/TransDecoder/TransDecoder/wiki) was used to identify potential coding sequence of the annotated genes. And the sequences of MSMs were aligned to the full-length coding sequence by BLAT (-t = dna -q = dna -oneOff = 2 -minIdentity = 80) to analyze the gene integrity.

All the transcripts were used for the subsequent analysis of Kyoto Encyclopediaof Genes and Genomes (KEGG). MARS model of DEGseq2 (Version: 1.4.5) (P-value = 1e-3, zscore = 4, q-value = 0.001, ThresholdKind = 5) was used to detected differentially expressed genes (DEGs). Sex-biased transcripts at each developmental stage were identified separately from DEGs under the conditions: $|\log_2\text{FoldChange}| \geq 1$ and FDR $\leq$ 0.05 and the final number of the sex biased transcripts were calculated by discarded the duplicated transcripts.

## Sequence subtraction

A set of subtraction approaches were used to identify the potential male specific transcripts as previously described [88–90] with some modifications. Two transcriptomes were used as a reference for transcriptome subtraction, including one female gonadal transcriptome of ten developmental stages from strain DA (S7 Table) and one gonadal transcriptome of females and temperature-dependent males (without MSMs) from strain A$^+$ (S10 Table). After transcriptome subtraction, the remaining transcripts were mapped to the female reference genome of hexaploid *C. gibelio* [34] for genome subtraction. The MSMs-linked transcripts were mapped to the transcriptomes and genome for subtraction using BLAT [87], and the sequences were discarded when the aligned length/full length is over 90% and the identity is higher than 98%. After subtraction, the remaining transcripts were aligned to the sequences of MSMs via BLAST with default parameters, and the well-aligned sequences of MSMs (identity>75% and aligned length $\geq$ 200 bp) were defined as gene fragments. Subsequently, these gene fragments were used for the second round of transcriptome/genome subtractions as described above. Gene fragments were annotated using the coding sequence of the female reference genome of hexaploid *C. gibelio* (GenBank accession: PRJNA546443), the non-redundant protein sequences database (https://www.ncbi.nlm.nih.gov), and the database of transcripts/splice variants (http://www.ensembl.org) orderly.

## RNA isolation and relative quantitative real-time PCR (qPCR)

Genotypic female and male gonads at 18, 22, 26, 30, and 35 dah were pooled respectively for total RNA isolation using SV Total RNA Isolation System (Promega). RNAs were reverse-transcribed via M-MLV reverse system, and qPCR was then performed on CFX96 Real-Time System (BioRad) with iQ SYBR Green Supermix (BioRad) as described previously [30]. *β-actin* was used as the internal reference. Each sample was analyzed in triplicates, and the relative expression level of target gene was calculated with $2^{-\Delta\Delta CT}$ method. The highest expression level of each gene fragment was used as control and defined as 1. PCR cycling conditions were: 95˚C for 1 min; 40 cycles of 15 s at 95˚C, 20 s at 58˚C, and 30 s at 72˚C in a 20 µl reaction mix. PCR primers were designed following the subsequent rules. If the MSM fragment had homolog sequences (identity > 75%) in the reference genome without MSMs, we designed the PCR primers according to the MSM fragment and made the 3' end nucleotide of each primer locate at the different sites between MSM fragment and homologous sequence of reference genome.

If the MSM fragment had no homolog sequence in the reference genome without MSM, we randomly designed the PCR primers according to the MSM fragment. Sequences of PCR primers are given in S11 Table.

## Supporting information

**S1 Fig. Schematic diagram of the relationship between male-specific sequence *Cg*-M-s and satellite repeat cluster *Cg-Ca*-CL1.** (A) *Cg*-M-s contains several intact and fragmental repeats of *Cg-Ca*-CL1. The sites of male-specific primers including *Cg*-M-s-F and *Cg*-M-s-R are marked by black arrows. (B) Sequence alignment between the consensus sequence of *Cg-Ca*-CL1 and the repeats of *Cg-Ca*-CL1 in *Cg*-M-s. (C) Peptide nucleic acid (PNA) probes used for fluorescence in situ hybridization (FISH). The sequences of PNA probes are indicated by yellow background. Each probe is labeled with three Cy3.
(TIF)

**S2 Fig. Co-localization of satellite repeat cluster *Cg-Ca*-CL1 and male-specific sequence *Cg*-M-s in *C. gibelio*.** (A-H) The *Cg-Ca*-CL1 probe and *Cg*-M-s probe were labeled with Biotin and Digoxin respectively, and red and green fluorescence were produced accordingly. FISH analysis was performed in metaphases of female *C. gibelio* (A-D) and male *C. gibelio* (E-H). Chromosomes were counterstained with DAPI and appeared blue. Scale bar = 5 μm.
(TIF)

**S3 Fig. FISH analysis and PCR detection of satellite repeat cluster *Cg-Ca*-CL1.** (**A, B**) FISH analysis of satellite repeat cluster *Cg-Ca*-CL1 (red) in metaphases of female *C. auratus* (A) and male *C. auratus* (B). (**C**) PCR assay of satellite repeat cluster *Cg-Ca*-CL1 in *C. gibelio* and *C. Carassius*. ♀, female; ♂, male. Scale bar = 5 μm.
(TIF)

**S4 Fig. Single chromosomal fluorescence microdissection.** (**A, C**) FISH detection on a male metaphase using PNA probes before (A) and after (C) microdissection. (**B, D**) The enlarged images of the white squares in (A) and (C), respectively. The red signals from PNA probes highlighted microchromosomes, and chromosomes were counterstained with DAPI and appeared blue. The white arrows indicate chromosome before (A and B) and after (C and D) isolation. Scale bar = 5 μm.
(TIF)

**S5 Fig. PCR detection of the male-specific marker.** (**A-C**) Male-specific marker in 8 randomly-picked males and 8 randomly-picked females in the offspring as well as the parental individuals from family 1 (A), family 2 (B), and family 3 (C). ♀, female; ♂, male; M, maternal individual; P, paternal individual.
(TIF)

**S6 Fig. Assessments of MSM sequence assemblies.** (**A, B**) Number (A) and total length (B) of aligned blocks referring to the female genome. (**C, D**) Number (C) and total length (D) of aligned blocks referring to the full-length transcriptomes. Different colors represent different lengths of aligned blocks. The X axis represents different datasets including clean reads, corrected reads, contigs assembled by CANU, contigs assembled by SPAdes, and contigs assemble by SMARTdenovo.
(TIF)

**S7 Fig. Analysis of gene content.** (**A**) Gene content of three MSMs. (B) Number of clean reads containing gene sequences. The X axis represents the proportion of gene sequences. The

Y axis indicates the number of clean reads.
(TIF)

**S8 Fig. qPCR analysis of gene fragments.** qPCR detection of gene fragments at early gonadal developmental stages including 18, 22, 26, 30, and 35 dah (days after hatch). The X axis represents the stages of gonad development. The Y axis represents the relative expression, and the highest expression level of each gene fragment was used as control and defined as 1. F, female; M, male.
(TIF)

**S9 Fig. Schematic diagrams of the family establishment.** (**A**) Establishment of a family containing a high proportion of male offspring. (**B**) Establishment of a family with all-female offspring. ♀, female; ♂, male; (+), with the male-specific genetic marker; (-) without the male-specific genetic marker.
(TIF)

**S1 Table. The consensus sequence of satellite DNA *Cg-Ca*-CL1.**
(DOCX)

**S2 Table. Sequencing summary of male-specific microchromosomes.**
(DOCX)

**S3 Table. The summary of the corrected sequences of MSMs.**
(DOCX)

**S4 Table. The summary of sequence assembly of MSMs by CANU.**
(DOCX)

**S5 Table. The summary of sequence assembly of MSMs by SPAdes.**
(DOCX)

**S6 Table. The summary of sequence assembly of MSMs by SMARTdenovo.**
(DOCX)

**S7 Table. PacBio sequencing summary of female and male gonads.**
(DOCX)

**S8 Table. Illumina sequencing summary of female and male gonads.**
(DOCX)

**S9 Table. Summary of 42 unique potential male-specific gene fragments. #** Red color indicates the gene fragments with a conserved coding sequence.
(DOCX)

**S10 Table. PacBio sequencing summary of female and male gonads without MSMs.**
(DOCX)

**S11 Table. Primers that used in this study.**
(DOCX)

## Acknowledgments

We thank Fang Zhou and Yan Wang (the Center for Instrumental Analysis and Metrology, Institute of Hydrobiology, Chinese Academy of Science) for providing confocal and microdissection services.

## Author Contributions

**Conceptualization:** Xi-Yin Li, Jian-Fang Gui.

**Formal analysis:** Miao Ding, Zhi-Xuan Zhu, Jun-Hui Chen, Meng Lu, Qian Shi, Yang Wang, Zhi Li, Xin Zhao, Tao Wang, Wen-Xuan Du, Chun Miao, Tian-Zi Yao, Ming-Tao Wang, Xiao-Juan Zhang, Zhong-Wei Wang, Li Zhou.

**Funding acquisition:** Xi-Yin Li, Li Zhou, Jian-Fang Gui.

**Investigation:** Zhi-Xuan Zhu, Jun-Hui Chen, Meng Lu, Qian Shi, Yang Wang, Zhi Li, Xin Zhao, Tao Wang, Wen-Xuan Du, Chun Miao, Tian-Zi Yao, Ming-Tao Wang, Xiao-Juan Zhang, Zhong-Wei Wang, Li Zhou.

**Methodology:** Miao Ding.

**Resources:** Miao Ding, Xi-Yin Li.

**Writing – original draft:** Xi-Yin Li.

**Writing – review & editing:** Xi-Yin Li, Jian-Fang Gui.

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
