## [Decision Letter · Decision Letter 0]

15 Mar 2021

Dear Dr Jian Fang Gui,

Thank you very much for submitting your Research Article entitled 'Genomic anatomy of male-specific microchromosomes reveals an evolutionary mechanism of sex chromosome in a gynogenetic fish' to PLOS Genetics.

The manuscript was fully evaluated at the editorial level and by independent peer reviewers. The reviewers appreciated the attention to an important problem, but raised many substantial concerns about the current manuscript. Based on the reviews, we will not be able to accept this version of the manuscript, but we would be willing to review a much-revised version. We cannot, of course, promise publication at that time.

Should you decide to revise the manuscript for further consideration here, your revisions have to address all specific points made by each reviewer. We will also require a detailed list of your responses to the review comments and a description of the changes you have made in the manuscript.

If you decide to revise the manuscript for further consideration at PLOS Genetics, please aim to resubmit within the next 90 days, unless it will take extra time to address the concerns of the reviewers, in which case we would appreciate an expected resubmission date by email to plosgenetics@plos.org.

[LINK]

We are sorry that we cannot be more positive about your manuscript at this stage. Please do not hesitate to contact us if you have any concerns or questions.

Yours sincerely,

Manfred Schartl

Guest Editor

PLOS Genetics

Gregory P. Copenhaver

Editor-in-Chief

PLOS Genetics

Reviewer's Responses to Questions

**Comments to the Authors:**

Reviewer #1: I was turned off by the very first sentence of this paper, and suspected the entire premise was wrong. I was forced to read, with considerable skepticism, the many previous papers by these authors. I was eventually convinced that they have identified some microchromosomes that contribute to male sex determination in this gynogenetic species. I have yet to see evidence that any sexual reproduction is occurring in this system. I believe these microchromosomes are simply a genomic parasite on the gynogenetic system. This is of course interesting, but represents a very different spin on the interpretation of the results.

Abstract

There is so much wrong with this first sentence. What evidence is there that ‘unisexual’ taxa evolve (not develop!) into sexual organisms? I was under the impression that gynogenetic taxa evolve from sexual (or dioecious) taxa, and eventually go extinct. It is entirely inappropriate to describe any of this as a “long-term evolutionary strategy”.

Introduction

Avise (ref 3) DOES NOT suggest that gynogenetic forms evolve into sexual species: “In summation, molecular, as well as ecological and distributional, data suggest that unisexual clonality in vertebrate animals can best be viewed as a genetic operation that sometimes offers a highly successful tactic in the ecological short term but that almost invariably fails as a long-term evolutionary strategy.”

Results

Figure 1A – Why are the two clouds of dots different colors (black and blue)?

Figure 2E-G – What exactly am I supposed to conclude from these figures? I do not see evidence of colocalization of green and red signals.

Discussion

Line 326 should also cite Clark (2019) for example of B chr evolving into sex chromosome. Clark FE, Kocher TD. Changing sex for selfish gain: B chromosomes of Lake Malawi cichlid fish. Sci Rep. 2019 Dec 27;9(1):20213

I think you should be more explicit about a model for this process. Gynogenetic carp seem to occasionally pick up DNA/chromosomes from the sperm donor. It appears that you may have identified a selfish supernumerary chromosome(s) that has exploited this process for its own reproduction. It may further carry a male sex determiner to further promote its transmission. It is still surprising to me that this seems to require the transmission of 3 microchromosomes. I would have expected the required elements to be closely linked on a single chromosome. But perhaps there is a mechanism that maintains linkage among the three microchromosomes. I do not see any evidence to support your idea (Figure 7) that these microchromosomes arose from with the hexaploidy gynogen. Instead, I suspect they arose from sperm DNA. Perhaps you can lay out several explicit hypotheses for the origins of these microchromosomes and evaluate their likelihood using all the sources of evidence available.

The analysis of repeat/gene content on the microchromosomes was a necessary exercise, but does not seem to have resulted in the identification of any particularly interesting candidates.

Materials and methods

How many fish were sampled in this study?

For the analysis in Figure 2C-D it appears to be 1 male (13 mChr) and 1 female (9 mChr)?

It is not clear there was any replication for Figure 2E-G.

The actual number of fish examined for each portion of the study needs to be made explicit.

Reviewer #2: Review uploaded as attachment

Reviewer #3: Ding and coauthors continued with the analysis of gibel carps’s male extra microchromosomes. Microchromosomes were also reported in other fish species and most of these microchromosomes were considered to be supernumerary chromosomes, which were suggested to have an effect on the phenotype of host species, especially the sex. In order to check how these extra microchromosomes are related to male sex determination in gibel carp, the authors applied chromosome microdissection, whole genome amplificatio, next-generation sequencing and bioinformatic analysis to unravel their DNA composition.

Considering that the copy number of supernumerary chromosomes often exceeds two, I’m wondering whether the three analysed MSMs represent different types of MSMs or only one type. Taken the highly similar sequence composition (shown in figures 3 and 4) it seems to me that only one type of MSM exists in gibel carp. Small sequence differences between the three next generation sequence sets are likely caused by the fact that MDA-based DNA amplification of a single chromosome never amplifies 100% of their DNA. The technical limitations of single-copy chromosome microdissection/sequencing is likely also the reason why only three and not four MSMs showed a male-specific PCR product (in figure 2G). I suggest providing experimental evidence that three types of MSMs exist or rewriting the manuscript accordingly.

The conclusion that MSMs accumulated mitochondria-derived DNA needs additional proof. Could it be a byproduct of the microisolation process? To provide proof, I would suggest a PCR experiment with primers specific for MSMs located mitochondrial DNA using genomic DNA of a female (negative control) and a male (positive control). Alternatively, FISH could be performed using labelled mitochondrial DNA. MSMs and mitochondria located FISH signals would proof that MSMs accumulated organellar DNA.

Page 5, lines 129-131

“In this study, we developed….”. This sentence is difficult to follow. I would suggest the following: “In this study, we analysed the sequence composition of microdissected FISH-labelled microchromosomes and identified three male-specific microchromosomes (MSMs) in hexaploidy C. gibelio.

Page 7, 184 – 186

It is unclear to me whether the 13 microchromosomes in a male and 9 microchromsomes in a female were isolated each from one metaphase cell each. Or did you isolate the mircochromosomes from several male and female cells.

**Have all data underlying the figures and results presented in the manuscript been provided?**

Reviewer #1: Yes

Reviewer #2: Yes

Reviewer #3: Yes

PLOS authors have the option to publish the peer review history of their article (what does this mean?). If published, this will include your full peer review and any attached files.

Reviewer #1: No

Reviewer #2: No

Reviewer #3: **Yes: **Andreas Houben

---

## [Decision Letter · Decision Letter 1]

9 Jul 2021

Dear Dr. Gui,

Thank you very much for submitting your Research Article entitled 'Genomic anatomy of male-specific microchromosomes in a gynogenetic fish' to PLOS Genetics.

The manuscript was fully evaluated at the editorial level and by independent peer reviewers. The reviewers appreciated the attention to an important problem, but raised some substantial concerns about the current manuscript. Based on the reviews, we will not be able to accept this version of the manuscript, but we would be willing to review a much-revised version. We cannot, of course, promise publication at that time.

If you decide to revise the manuscript for further consideration at PLOS Genetics, please aim to resubmit within the next 60 days, unless it will take extra time to address the concerns of the reviewers, in which case we would appreciate an expected resubmission date by email to plosgenetics@plos.org.

[LINK]

We are sorry that we cannot be more positive about your manuscript at this stage. Please do not hesitate to contact us if you have any concerns or questions.

Yours sincerely,

Manfred Schartl

Guest Editor

PLOS Genetics

Gregory P. Copenhaver

Editor-in-Chief

PLOS Genetics

Reviewer's Responses to Questions

**Comments to the Authors:**

Reviewer #2: First, I would like to thanks the authors for the important modifications that have been made on their manuscript. Many of these modifications provide good and interesting answers to most of my comments.

However, I still have some concerns on the MSM gene annotation as it is unclear in this new revised manuscript what, and how many, genes are present on the MSMs with a potential open reading frame (intact genes). You are mentioning “gene fragments” but it would be important to know how many of them have a conserved CDS compared to their paralogous copy (or copies) in the A chromosome part of the genome. A more detailed annotation should be provided (not only a blast analysis).

It is also unclear for me how you designed primers to be specific or not of the MSMs fragments. This is important to evaluate if this is really a specific expression of the MSM genes.

The question of what do you called “clean reads” (see below) may be also important to understand and to evaluate what you have done on the assembly of MSMs.

More generally I think that your results are interesting and should deserve publication, but I would suggest again not to oversell your results with straightforward conclusions that are not well supported by your results. Being more cautious with conclusions will not decrease the quality of your results. I already mentioned that important point in my first review but there are still too many statements that are not well-supported by your results

Specific comments:

Line 31-32: “Here, we identify a massively expanded satellite DNA cluster on microchromosomes of gibel carp via all-to-all similarity comparison”. Alone without further explanation this sentence cannot be understood by readers who will only read your abstract. An all-to-all similarity comparison does not mean anything if not described with more details

Lines 57-58: “Our previous studies have revealed that the supernumerary microchromosomes play a male determination role in gibel carp” I do not think that you can say that. Your previous studies revealed a correlation between some supernumerary microchromosomes and the male phenotype which could suggest that they MAY play a role in male determination. Same comment throughout your manuscript like for example lines 123-125 ‘ effects on sex determination [31-34]. In hexaploid C. gibelio, supernumerary microchromosomes in males have also been found to play a male determination role [23, 35]”

Lines 62-66: “These findings indicate that the male-specific microchromosomes resembling common features of sex chromosomes may be the main driving forces for male occurrence in gynogenetic gibel carp, through which the gynogenetic gibel carp can conquer the disadvantages of unisexual reproduction with evolutionary long existence”.

The first part of the sentence is OK except that it is not explained clearly (first mention here) what are the resemblance with sex chromosomes. For the second part this is pure speculation (and this sentence is also present in the abstract, the end of the introduction, the discussion …) and should not be mentioned without a lot more caution.

Line 136: what do you mean exactly by “sex related gene accumulation » ?

Line 137: you did not provide any indication of a “and male determination function ». You could probably say that some genes on these B have a sex-specific or sex-biased expression profile and that’s all.

Lines 220-221: “All the contigs and corrected reads displayed much lower alignment level to the references in sharp contrast with the clean reads” what are clean reads? Isn’t it expected (or at least a possibility) to have B microchromosome reads remapping with a lower quality on a reference that do not have any B inside. If some B sequences have substantially evolved from their type A chromosome origin by only considering “clean reads” (if I am right in my own definition of your “clean reads”) you potentially removed a large part (potentially B-specific) of the B sequence.

Lines 229-231: “The total mapping rates to the reference genome of MSM 1, MSM 2, and MSM 3 were 0.46%, 0.13%, and 0.56%, respectively.” Does that mean that 1) only 0.46% pf MSM 1 are mapped on the reference genome OR 2) does it mean that 0.46% of the reference genome size is covered by MSM 1 reads? If 1) this is surprising given that you state that B are mainly made up from A regions. If 2) it needs to be better explained

Lines 322-323: “trpv4 (transient receptor potential cation channel, subfamily V, member 4), which was suggested to be a possible regulator of sox9 [42, 43] and play a male differentiation role in the American alligator”. This is not totally correct to state that trpv4 plays a male differentiation role in alligator. Its expression is indeed induced by temperatures proximate to the TSD-related temperature in alligators and its chemical disruption also disturbed the expression pattern of gonadal transcriptomes. These are interesting correlations but do not really prove that it plays a role. Just a matter of being cautious in the wording again.

Lines 359: what is the meaning of “variable” in “the evolution of variable sex chromosomes”?

Discussion: I found that the discussion on the evolutionary trajectory of the MSMs toward a future sex chromosome in a species that will ultimately have a transition from unisexual reproduction to sexual reproduction is pure speculation. Going back from now to the past is already not an easy exercise but projecting current results in the future is even more difficult. Personally, I would not discuss this.

Lines 401-402: I am sorry but you cannot state that “Our present results strongly indicate that MSMs are responsible for male determination in C. gibelio [18, 23, 36]”. This is not to be nasty but you should refrain from overselling your nice results. Just keep conclusions that strictly fit with your results and it will still be an interesting paper. In that case you just provide additional evidence that MSMs could be … There is no strong indication as you mentioned only (and that’s already good) the characterization of MSM genes (or gene fragments) with a male-specific or male-biased expression.

I am not a native English speaker but could you double check that the following sentences are correct? Many sentences of this manuscript look weird to me and a detailed revision of English may be need.

“… is predicted to be extinction … “?

Line 122: “many species [30], which indicate supernumerary chromosomes may have functional” = indicate that?

Lines 389-390: « to be evolved supernumerary chromosomes … » to be evolved from?

Reviewer #3: The authors improved the manuscript, but one question remained.

(Page 9, line 239 – 240) Considering that the sequence characterization of MSM1, MSM2 and MSM3 is based on the amplified DNA of single chromosomes, I strongly recommend toning down the statement “….We could deduce that the MSM 1 and MSM 3 had more similar sequence components, which were divergent from MSM 2.” I’m not aware of any WGA (including MDA-based) method which amplifies 1 to 1 the DNA of a microisolated chromosome. Very often, a bias is observed, which could easily explain the observed differences between MSM1, MSM2 and MSM3. Could it be in the end a single MSM chromosome type with many copies? Besides the DNA sequences of the miroisolated/amplified chromosomes, do you have any evidence (eg FISH data) that the MSMs differ in sequence composition from each other?

Minor points:

Abstract, line 31 – 37

Please use past tense.

Page 5, line 117

Rephrase. “..accumulate organelle genomes”. Suggestion: “...accumulate organelle genome-derived sequences.”.

Page 6, lines 154 – 160, Please rephrase, it is difficult to understand.

Page 7, line 187 and 17, line 454, Did you amplify the chromosomes individually?

If yes, rewrite like: ” ..and individually amplified microisolated chromosomes…”.

Page 34, line 1037

Do you mean: Mitotic cells of three males were used as replication for co-localization analysis?

**Have all data underlying the figures and results presented in the manuscript been provided?**

Reviewer #2: Yes

Reviewer #3: Yes

PLOS authors have the option to publish the peer review history of their article (what does this mean?). If published, this will include your full peer review and any attached files.

Reviewer #2: No

Reviewer #3: **Yes: **Andreas Houben

---

## [Decision Letter · Decision Letter 2]

9 Aug 2021

Dear Dr Gui,

We are pleased to inform you that your manuscript entitled "Genomic anatomy of male-specific microchromosomes in a gynogenetic fish" has been editorially accepted for publication in PLOS Genetics. Congratulations!

Yours sincerely,

Manfred Schartl

Guest Editor

PLOS Genetics

Gregory P. Copenhaver

Editor-in-Chief

PLOS Genetics

Comments from the reviewers (if applicable):

Reviewer's Responses to Questions

**Comments to the Authors:**

Reviewer #2: Thanks for this revision that answers all my previous comments

**Have all data underlying the figures and results presented in the manuscript been provided?**

Reviewer #2: Yes

PLOS authors have the option to publish the peer review history of their article (what does this mean?). If published, this will include your full peer review and any attached files.

Reviewer #2: No

**Data Deposition**

http://datadryad.org/submit?journalID=pgenetics&manu=PGENETICS-D-21-00204R2

**Press Queries**

---

## [Editor Report · Acceptance letter]

2 Sep 2021

PGENETICS-D-21-00204R2 

Genomic anatomy of male-specific microchromosomes in a gynogenetic fish 

Dear Dr Gui, 

We are pleased to inform you that your manuscript entitled "Genomic anatomy of male-specific microchromosomes in a gynogenetic fish" has been formally accepted for publication in PLOS Genetics! Your manuscript is now with our production department and you will be notified of the publication date in due course.

With kind regards,

Agnes Pap

PLOS Genetics

On behalf of:
